# A theory vade mecum for PSI experiments

G. Colangelo[1], F. Hagelstein[2], A. Signer[2,3]* and P. Stoffer[4]

**1** Albert Einstein Center for Fundamental Physics, Institute for Theoretical Physics,
University of Bern, Switzerland
**2** Paul Scherrer Institut, 5232 Villigen PSI, Switzerland
**3** University of Zurich, Physik-Institut, 8057 Zurich, Switzerland
**4** University of Vienna, Faculty of Physics, Boltzmanngasse 5, 1090 Vienna, Austria

* adrian.signer@psi.ch

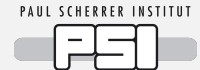

## Abstract

**This article gives a compact introduction and overview of the theory underlying the experiments described in the rest of this review.**

## 5.1 Introduction

The purpose of this article is to give a broad overview of the theory background to the experiments that have been and are carried out at the Paul Scherrer Institute. Space limitations make it impossible to go into depth or provide a self-contained theoretical summary. Much more modestly, we aim to put the experiments into context and provide key references for further reading. The experiments we refer to are listed in Table 5.1 and they will be described in greater detail in separate sections/articles of the Review of Particle Physics at PSI [1–23]. These experiments either lead to precise determinations of physical parameters required as input for other experiments (e.g., muon life time, pion mass), or search for physics beyond the Standard Model (BSM). The BSM searches proceed along different frontiers. One way to search for new physics is to consider physical observables whose Standard Model (SM) contributions either vanish or are too small to be experimentally accessible. In other words, they are identical to zero for practical purposes. Examples are charged lepton-flavor violating (cLFV) muon decays or a permanent neutron electric dipole moment (EDM). To put constraints on the branching ratios of BSM decays, one has to observe a large number of decays. This is, thus, called a search at the intensity frontier. Another way to search for new physics is to consider precision observables and search for deviations from the SM expectations. Prominent examples are the precision QED tests with muonium, as well as the precision laser spectroscopy experiments with muonic atoms. These are, thus, called searches at the precision frontier.

The low-energy experiments at PSI are complementary to the experiments at LHC, which sit at the energy frontier.

After a general overview of the theoretical methods applied to describe the processes and bound states in Table 5.1, we will, in turn, consider the muon, the proton, nucleons and nuclei, the free neutron, and the pions.

## 5.2 Overview

The experiments we are primarily concerned with involve low-energy interactions of electrons, muons, protons, neutrons, and pions. In Section 5.2.1 we first describe these interactions in the SM before we discuss the generalization to BSM scenarios in Section 5.2.2. While the theoretical methods for these cases are dominated by perturbative expansions in the couplings, Section 5.2.3 is devoted to hadronic effects that often play an important part in low-energy experiments.

### 5.2.1 Standard Model at low energies

In the SM the dynamics of the particles listed above is described by the gauge theory of strong and electroweak interactions. In view of the large masses of the Higgs and weak gauge bosons, the weak part of the SM Lagrangian is essentially frozen at low energies (it will later be considered as a small correction). In this regime, the SM reduces to the standard QED and QCD Lagrangian

$$\mathcal{L}_{\text{QED+QCD}} = \sum_f \bar{f}\left(i\slashed{D} - m_f\right)f - \frac{1}{4}F_{\alpha\beta}F^{\alpha\beta} - \frac{1}{4}G_{\alpha\beta}G^{\alpha\beta}, \tag{5.1}$$

where the electromagnetic and gluonic field-strength tensors are expressed in terms of the photon and gluon fields, $A^\alpha$ and $G^\alpha$, as $F^{\alpha\beta} = \partial^\alpha A^\beta - \partial^\beta A^\alpha$, $G^{\alpha\beta} = \partial^\alpha G^\beta - \partial^\beta G^\alpha - ig_s[G^\alpha, G^\beta]$, and where for clarity we have omitted gauge-fixing and ghost terms. The sum runs over all fermions of mass $m_f$, electric charge $eQ_f$, and color charge $g_s t_f^a$, and the covariant derivative acts on the fermion fields as $D_\alpha f = (\partial_\alpha - ieQ_f A_\alpha - ig_s t_f^a G_\alpha^a)f$. For $f = \ell \in \{e, \mu, \tau\}$ we have $Q_\ell = -1$ and $t_\ell^a = 0$, whereas for quarks $Q_u = 2/3$, $Q_d = -1/3$, and $t_{u,d}^a = \lambda^a/2$ with Gell-Mann matrices $\lambda^a$. In several experiments of interest here the photon acts as a probe: it is coupled to the electromagnetic current $J_{\text{em}}^\alpha$ as

$$\mathcal{L}_{\text{QED}}^{\text{int}} = eA_\alpha J_{\text{em}}^\alpha \equiv eA_\alpha \sum_f Q_f \bar{f}\gamma^\alpha f. \tag{5.2}$$

If we use (5.1) to compute the matrix element of $J_{\text{em}}^\alpha$ between two states of pointlike leptons $\ell$ with momenta $p_1$ and $p_2 = p_1 + q$, we find

$$\langle \ell(p_2)|J_{\text{em}}^\alpha|\ell(p_1)\rangle = \bar{u}(p_2, m_\ell)\left(F_1^{(\ell)}(q^2)\gamma^\alpha + F_2^{(\ell)}(q^2)\frac{i\sigma^{\alpha\beta}q_\beta}{2m_\ell}\right)u(p_1, m_\ell), \tag{5.3}$$

where $u$ and $\bar{u}$ are the usual spinors. The decomposition (5.3) directly follows from the Lorentz and $U(1)_{\text{em}}$ gauge symmetries of the theory and is valid beyond perturbation theory. While $F_1^{(\ell)}$ is related to the electric charge, $F_2^{(\ell)}$ is related to the anomalous magnetic moment (AMM) of $\ell$ as

$$F_2^{(\ell)}(0) = a_\ell = \frac{(g-2)_\ell}{2}. \tag{5.4}$$

Table 5.1: Processes and particles (bound states) that are investigated at PSI, where the driving interaction to be studied is indicated by the color as follows: BSM, weak, weak and try to learn about strong, EM, EM and try to learn about strong, strong. In addition the mass or charge radius of particles are measured. The section number refers to the Review of Particle Physics at PSI.

| | experiment | section | process / particles / (bound states) |
|---|---|---|---|
| [1] | muon decay | 6 | $\mu^+ \to e^+ \nu_e \bar{\nu}_\mu$ |
| [2] | MuLan | 16 | $\mu^+ \to e^+ \nu_e \bar{\nu}_\mu$ |
| [3] | SINDRUM | 7 | $\mu^+ \to e^+ ee$, $\mu^+ \to e^+ \nu_e \bar{\nu}_\mu ee$, $\pi^+ \to e^+ \nu_e ee$, $\pi^0 \to ee$ |
| [4] | SINDRUM II | 8 | $\mu^- {}^A_Z N \to e^- {}^A_Z N$    for Au, Pb, Ti |
| [5] | MEG | 19 | $\mu^+ \to e^+ \gamma$, $\mu^+ \to e^+ \nu_e \bar{\nu}_\mu \gamma$, $\mu^+ \to e^+ X \to e^+ \gamma\gamma$ |
| [6] | Mu3e | 20 | $\mu^+ \to e^+ ee$, $\mu^+ \to e^+ \nu_e \bar{\nu}_\mu ee$ |
| [7] | Mspec, Mu-Mass | 29 | $M = (\mu^+ e^-)$, $\mu^+$ |
| [8] | MACS | 9 | $M = (\mu^+ e^-) \leftrightarrow \bar{M} = (\mu^- e^+)$ |
| [9] | CREMA | 21 | $(\mu^- p)$, $(\mu^- d)$, $(\mu^- \text{He})$,    $p$, $d$, He |
| [10] | muX | 22 | $(\mu^- {}^A_Z N)$,    ${}^{248}_{96}\text{Cm}$, ${}^{226}_{88}\text{Ra}$ |
| [11] | MUSE | 23 | $e^\pm p \to e^\pm p$, $\mu^\pm p \to \mu^\pm p$ |
| [12] | MuCap | 17 | $\mu^- p \to \nu_\mu n$ |
| [13] | MuSun | 18 | $\mu^- d \to \nu_\mu nn$ |
| [14] | pionic hydrogen | 14 | $(\pi^- p)$, $(\pi^- d)$ |
| [15] | pionic helium | 26 | $(\pi^- e^- \ {}^4\text{He}^{++})$, $\pi^-$ |
| [16] | nTRV | 15 | $n \to p e^- \bar{\nu}_e$ |
| [17] | nEDM | 27 | $n$, $n$ |
| [18] | indirect nEDM | 28 | $n$ / dark matter / exotic |
| [19] | negative pions | 10 | $(\pi^- p)$, $\pi^-$ |
| [20] | positive pions | 11 | $\pi^+ \to \mu^+ \nu_\mu$, $\pi^+$, $\nu_\mu$ |
| [21] | neutral pions | 12 | $\pi^- p \to \pi^0 n$, $\pi^0$ |
| [22] | PiBeta | 24 | $\pi^+ \to \pi^0 e^+ \nu_e$, $\pi^+ \to e^+ \nu_e (+\gamma)$, $\mu^+ \to e^+ \nu_e \bar{\nu}_\mu \gamma$ |
| [23] | PEN | 25 | $\pi^+ \to e^+ \nu_e (+\gamma)$, $\mu^+ \to e^+ \nu_e \bar{\nu}_\mu \gamma$ |

In contrast to the leptons, quarks do not appear as free particles in nature, but are confined inside hadrons by the strong interaction. The general principles on which the decomposition (5.3) is based, also hold for non-pointlike particles, such as the nucleons $N \in \{p, n\}$

$$\langle N(p_2)|J_{\text{em}}^\alpha|N(p_1)\rangle = \bar{u}(p_2, m_N)\left(F_1^{(N)}(Q^2)\gamma^\alpha + F_2^{(N)}(Q^2)\frac{i\sigma^{\alpha\beta}q_\beta}{2m_N}\right)u(p_1, m_N), \qquad (5.5)$$

where we have introduced the common definition $Q^2 \equiv -q^2$. A relation between the AMM and $F_2^{(N)}$ analogous to (5.4) still holds. However, this quantity depends on strong dynamics, which at low energies cannot be computed in perturbation theory.

In the case of the nucleons, often the electric and magnetic form factors

$$G_E^{(N)}(Q^2) \equiv F_1^{(N)}(Q^2) - \frac{Q^2}{4m_N^2}F_2^{(N)}(Q^2), \qquad G_M^{(N)}(Q^2) \equiv F_1^{(N)}(Q^2) + F_2^{(N)}(Q^2), \qquad (5.6)$$

are used. In the limit of small $Q^2$ all form factors $F_i(Q^2)$ can be understood as the Fourier transform of an extended classical 'charge' distribution $\rho_i(r)$ in the Breit frame where $q^\mu = (0, \vec{q})$. Upon expansion in small $Q^2$ we get

$$F_i(Q^2) = \int d^3\vec{r}\, e^{-i\vec{q}\cdot\vec{r}}\,\rho_i(r) = \int d^3\vec{r}\,\rho_i(r) - \frac{1}{6}Q^2\int d^3\vec{r}\, r^2\,\rho_i(r) + \dots \qquad (5.7)$$

This leads to a general expression for the second moment of the charge distribution $\rho_i$

$$r_i^2 \equiv \frac{1}{N}\int d^3\vec{r}\, r^2\,\rho_i(r) = -6\frac{1}{N}\frac{dF_i(Q^2)}{dQ^2}\bigg|_{Q^2=0}, \qquad N = \begin{cases} 1 & \text{if } F_i(0) = 0, \\ F_i(0) & \text{else.} \end{cases} \qquad (5.8)$$

The relation above is used for example to determine the root-mean-square, $R_i = \sqrt{r_i^2}$, charge and magnetic radii of the proton as well as the axial radius of the nucleon.

If we now consider the weak interactions, we must arrange fermions into left-handed doublets and right-handed singlets. An important role for low-energy processes is played by the charged weak current

$$J_{\text{cc}}^\alpha = \sum_\ell \bar{\nu}_\ell \gamma^\alpha P_L \ell + \sum_{ij} V_{ij}\bar{u}_i \gamma^\alpha P_L d_j, \qquad (5.9)$$

which couples only to left-handed fermions, $P_L \equiv (1 - \gamma_5)/2$. In the sum over the quark-field terms, the CKM matrix $V_{ij}$ describes the flavor-changing effects of the weak interactions. Including for completeness also the neutral weak current $J_{\text{nc}}^\alpha$, the interactions of (5.2) are modified to

$$\mathcal{L}_{\text{EW}}^{\text{int}} = eA_\alpha J_{\text{em}}^\alpha + \frac{g}{\sqrt{2}}\left(W_\alpha^+ J_{\text{cc}}^\alpha + \text{h.c.}\right) + g_Z Z_\alpha J_{\text{nc}}^\alpha, \qquad (5.10)$$

where $g = e/\sin\theta_W$, $g_Z = g/\cos\theta_W$ are the weak $SU(2)_L$ couplings that can be expressed in terms of $e$ and the electroweak mixing (Weinberg) angle $\theta_W$. At the typical energy of processes considered here, much smaller than $m_W$ and $m_Z$, the $W$ and $Z$ boson masses, we can integrate out the $W$ and $Z$ bosons and adopt an effective field theory (EFT) approach. This results in the Fermi theory of current-current interactions

$$\mathcal{L}_{4F} = -\frac{4G_F}{\sqrt{2}}\left(J_{\text{cc}}^\alpha(J_{\text{cc}})_\alpha^\dagger + J_{\text{nc}}^\alpha(J_{\text{nc}})_\alpha\right), \qquad (5.11)$$

where $4G_F/\sqrt{2} = g^2/(2m_W^2)$ is the matching (Wilson) coefficient at tree level. Using (5.9) (and the corresponding expression for $J_{\text{nc}}^\alpha$) to express $\mathcal{L}_{4F}$ in terms of fermion fields we end

up with vector contact interactions. They correspond to dimension-6 four-fermion vector operators of the generic form

$$\left[O_{\{\ell/q\}}^{V,XY}\right]_{ijkl} = \left(\bar{\psi}_i \gamma^\alpha P_X \psi_j\right)\left(\bar{\psi}_k \gamma_\alpha P_Y \psi_l\right), \tag{5.12}$$

where $X, Y \in \{L, R\}$ and $\{i, j, k, l\}$ are generation indices. The notion 'vector' refers to the Lorentz structure of the bilinears, which in turn is closely related to the nature of the exchange particle that is integrated out. Since the fermion fields $\psi_i$ can be quarks or leptons of any generation, there are in principle quite a lot of different operators. However, only a subset of those are generated by integrating out the $W$ and $Z$ fields. In particular, there are no charged cLFV operators due to an accidental symmetry of the SM.

Because the masses of the top quark and the Higgs boson are of the same order as $m_W$, these fields can also be integrated out. Operators beyond the four-fermion vector operators appear in the SM with an additional suppression, such as scalar dimension-6 four-fermion operators

$$\left[O_{\{\ell/q\}}^{S,XY}\right]_{ijkl} = \left(\bar{\psi}_i P_X \psi_j\right)\left(\bar{\psi}_k P_Y \psi_l\right), \quad X, Y \in \{L, R\}, \tag{5.13}$$

which are parametrically suppressed by Yukawa couplings [24], or dimension-5 dipole operators (and their Hermitian conjugate)

$$\left[O_{\{\ell/q\}\gamma}^{D}\right]_{ij} = \left(\bar{\psi}_i \sigma_{\alpha\beta} P_R \psi_j\right) F^{\alpha\beta}, \quad \left[O_{qG}^{D}\right]_{ij} = \left(\bar{\psi}_i \sigma_{\alpha\beta} G^{\alpha\beta} P_R \psi_j\right), \tag{5.14}$$

which appear at the loop level. Thus, we arrive at an EFT that consistently describes low-energy processes. It only contains fields with masses much lower than $m_W$. In particular, the photon and the gluons are the only gauge bosons present. The gauge symmetry of the SM, $SU(3)_c \times SU(2)_L \times U(1)_Y$, is reduced to the gauge symmetry of QCD and QED, $SU(3)_c \times U(1)_{\text{em}}$. The effect of the heavy degrees of freedom of the SM is encoded in the Wilson coefficients that multiply the operators, with $G_F$ in (5.11) being one such example.

### 5.2.2 Low-energy physics beyond the Standard Model

Many of the experiments listed in Table 5.1 are motivated by the search for new physics. One can think of a plethora of BSM scenarios. They rely on different interaction mechanisms, and can be roughly classified based on the masses of the BSM particles and their coupling strengths.

Light BSM particles should only have a small coupling to SM particles, which would explain their small contribution to physical observables. The most prominent examples are dark photons, axions, or axion-like particles (ALPs). The axion has been proposed as a dynamical solution to the strong CP problem [25–28], i.e., the "naturalness" problem of the small QCD $\theta$ parameter. It is introduced as the Nambu-Goldstone boson associated with a spontaneously broken additional global $U(1)_{\text{PQ}}$ symmetry of the SM Lagrangian. The modified SM Lagrangian reads

$$\mathcal{L}_{\text{SM}}^{\text{eff.}} = \mathcal{L}_{\text{SM}} + \mathcal{L}_{\text{int}}[\partial^\mu a_{\text{phys.}}/f_a; \psi] \tag{5.15}$$

$$- \frac{1}{2}\partial^\mu a_{\text{phys.}} \partial_\mu a_{\text{phys.}} - \frac{m_a^2}{2} a_{\text{phys.}}^2 + \frac{a_{\text{phys.}}}{f_a} \zeta \frac{g_s^2}{32\pi^2} \tilde{G}_{\alpha\beta} G^{\alpha\beta},$$

where $a_{\text{phys.}} = a - \langle a \rangle$ is the physical axion field with mass $m_a$, and $f_a$ is the $U(1)_{\text{PQ}}$ symmetry breaking scale. The axion is a pseudoscalar that couples derivatively to any field $\psi$. In addition, because of the chiral anomaly of the $U(1)_{\text{PQ}}$ current, it directly couples to the gluon density, where $\zeta$ is a model-dependent parameter. The minimum of the effective potential occurs at the axion vacuum expectation value $\langle a \rangle = -\theta f_a/\zeta$, which leads to a cancellation

of the CP violating QCD $\theta$ term and dynamically solves the strong CP problem. The defining characteristic of the axion, distinguishing it from an ALP, is $m_a f_a \sim m_\pi f_\pi$. This follows from mixing of the axion with the light $\pi$ and $\eta$ mesons.

In the following, we will be mainly concerned with heavy BSM particles. In Section 5.2.1, we described how the $W$ and $Z$ bosons can be integrated out in an EFT approach. Similarly, whatever BSM physics there is, as long as it respects QED and QCD gauge symmetry and involves degrees of freedom with a 'large' mass scale $\Lambda$, it can be integrated out and its effects will be encoded in Wilson coefficients of gauge-invariant higher-dimensional operators. Operators that were absent in the SM case might now be generated. Thus, we are led to write down the most general relativistic Lagrangian that respects electromagnetic $U(1)_{\mathrm{em}}$ and strong $SU(3)_c$ gauge invariance and obtain a general low-energy effective field theory (LEFT)

$$\mathcal{L}_{\mathrm{LEFT}} = \mathcal{L}_{\mathrm{QED+QCD}} + \frac{1}{\Lambda}\sum_i C_i^{(5)} O_i^{(5)} + \frac{1}{\Lambda^2}\sum_j C_j^{(6)} O_j^{(6)} + \dots \tag{5.16}$$

Here $\Lambda$ is the scale of physics that is not dynamically described by the degrees of freedom present in $\mathcal{L}_{\mathrm{LEFT}}$. If we include all charged leptons and all quarks apart from the top in $\mathcal{L}_{\mathrm{LEFT}}$, the scale $\Lambda$ is assumed to be larger than the mass of the $b$ quark but not larger than the electroweak scale $m_W$. The sums $i$ and $j$ run over all possible operators of dimension 5 and 6, respectively. Typically, operators of dimension larger than 6 are neglected. $O^{(5)}$ and $O^{(6)}$ denote the operators, $C^{(5)}$ and $C^{(6)}$ are the corresponding Wilson coefficients. Operators that are related through Fierz identities or those that can be eliminated through equations of motion are not included. Naturally, the choice of the operator basis is not unique, but a complete basis up to dimension 6 can be found in [24].

The Lagrangian (5.16) provides a consistent quantum-field theoretical framework to relate low-energy measurements to the determination of parameters of the SM and constraints on BSM physics. Many different routes have been taken to generically parametrize low-energy observables and measuring or constraining the associated parameters. The prime example is the Michel decay, where an analysis with initially a single parameter [29] was generalized and written in terms of parameters related to scalar, vector and tensor contact interactions[1] [30]. A similar effort has been made for cLFV decays $\mu \to e\gamma$ and $\mu \to eee$ considering lepton-flavor-violating contact interactions [31].

At first sight this is very similar to constraining the Wilson coefficients of (5.16). Indeed, the bulk of the operators of (5.16) are also scalar, vector and tensor interactions. However, the Wilson coefficients are well-defined couplings of a quantum field theory. In particular, typically they run and mix under renormalization-group evolution (RGE). If a low-energy observable is expressed in terms of Wilson coefficients, they are understood to be evaluated at the low scale, $C_i^{(n)}(m_\mu)$. On the other hand, to relate the Wilson coefficients of the EFT to a BSM model, the heavy degrees of freedom of the latter have to be integrated out. This yields the Wilson coefficients at the high scale, $C_i^{(n)}(\Lambda)$. Including RGE of $C_i^{(n)}(\Lambda)$ to $C_i^{(n)}(m_\mu)$ is not in the first instance about increasing precision, but to include qualitatively new effects through mixing. This has a profound impact on using low-energy measurements to constrain BSM models.

Of course, it is also possible that BSM physics appears only at a scale much larger than $m_W$. If this is the case, in a first step another effective theory has to be used, the SM effective field theory (SMEFT). This is a theory similar to (5.16), but with all fields and symmetries of the SM. It contains all operators $\mathcal{O}_i^{(n)}$ expressed in terms of the SM gauge fields, the Higgs doublet, as well as left-handed doublet and right-handed singlet fermion fields that respect

---

[1] Section 6: Muon decay [1].

the SM gauge symmetry $SU(3)_c \times SU(2)_L \times U(1)_Y$,

$$\mathcal{L}_{\text{SMEFT}} = \mathcal{L}_{\text{SM}} + \frac{1}{\Lambda}\big(\mathcal{C}^{(5)}\mathcal{O}^{(5)} + \text{h.c.}\big) + \frac{1}{\Lambda^2}\sum_j \mathcal{C}_j^{(6)}\mathcal{O}_j^{(6)} + \dots \tag{5.17}$$

SMEFT has only one dimension-5 operator $\mathcal{O}^{(5)}$ (and its Hermitian conjugate). This is the Weinberg operator [32] that is associated with neutrino masses. At dimension 6 there are numerous operators, some of which violate baryon number. As for $\mathcal{L}_{\text{LEFT}}$ different bases are possible, but the so-called Warsaw basis [33] is used frequently.

In the case $\Lambda \gg m_W$ the input of the BSM model is given through Wilson coefficients $\mathcal{C}_i^{(n)}(\Lambda)$. Then, the RGE is used to obtain $\mathcal{C}_i^{(n)}(m_W)$. In a next step, SMEFT is matched to LEFT at the electroweak scale. This means that $C_i^{(n)}(m_W)$ are expressed in terms of $\mathcal{C}_i^{(n)}(m_W)$. Finally, the Wilson coefficients of LEFT, $C_i^{(n)}(m_W)$, are run with the RGE of LEFT from the scale $m_W$ to the low scale $m_\mu$, and we are ready to express physical low-energy observables. The complete dimension-6 RGEs of SMEFT and LEFT, and the matching equations between the two EFTs are known at one loop [34–38], whereas beyond only partial results are known.

Now that we have a framework that incorporates the effects of the full SM and potential BSM physics on low-energy observables, we can return to our starting point, the matrix elements of the electromagnetic currents. Moving from (5.1) to (5.16) leads to a generalization of (5.2), (5.3), and (5.5). In particular, the current itself is modified and includes additional terms from the dimension-5 dipole operators. The most general expression for a vector current depending on $p_1$ and $p_2$ can be written as combination of six possible structures: $\gamma^\alpha$, $\gamma^\alpha\gamma_5$, $q^\alpha$, $q^\alpha\gamma_5$, $q_\beta\sigma^{\alpha\beta}$ and $q_\beta\sigma^{\alpha\beta}\gamma_5$. Replacing $q = p_2 - p_1$ by $p_2 + p_1$ does not lead to new independent structures, as can be shown by using the Dirac equation. Since the electromagnetic current is conserved $\partial_\alpha J_{\text{em}}^\alpha = 0$ only four terms remain and we get

$$\langle f(p_2)|J_{\text{em}}^\alpha|f(p_1)\rangle = \bar{u}(p_2, m_f)\Big(F_1^{(f)}(q^2)\gamma^\alpha + \big(F_2^{(f)}(q^2) - i\gamma_5 F_3^{(f)}(q^2)\big)\frac{i\sigma^{\alpha\beta}q_\beta}{2m_f} \tag{5.18}$$
$$+ F_4^{(f)}(q^2)\frac{1}{m_f^2}\big(q^2\gamma^\alpha - 2m_f q^\alpha\big)\gamma_5\Big)u(p_1, m_f).$$

The CP-violating form factor $F_3$ is associated with the EDM of the lepton $d_f$ through

$$d_f = \frac{eF_3^{(f)}(0)}{2m_f}. \tag{5.19}$$

In the SM, $d_f$ starts to receive contributions at three loops for quarks [39] and at four loops for leptons [40], induced by the CP violation in the CKM matrix. For protons and neutrons there is an additional source for an EDM [41] through the CP-violating $\theta$ term in QCD

$$\mathcal{L}_{\text{QCD}} \supset \frac{g_s^2\theta}{32\pi^2}\tilde{G}_{\alpha\beta}G^{\alpha\beta}, \tag{5.20}$$

which we have neglected in (5.1). This term has to be included as it respects $SU(3)_c$ gauge invariance. Even though it can be written as a total derivative and, so does not affect the classical equations of motion, the $\theta$ term does have effects at the quantum level. Thus strong interactions seem to violate CP. However, due to experimental constraints on the neutron EDM, we know that the $\theta$ parameter is extremely small, see Section 5.6. The lack of an explanation for this smallness is referred to as the strong CP problem. In generic BSM models, one usually expects much larger CP-violating effects [42,43]. The parity-violating anapole form factor $F_4$ is

also induced due to weak interactions of the SM, or potentially through BSM effects. However, it is not an observable by itself [44].

As mentioned above, matrix elements of the weak charged current $J_{cc}^\alpha$ also play an important role. It gives rise to non-vanishing matrix elements between different particles of left-handed $SU(2)$ doublets, such as $(\nu_\ell, \ell)$ or $(p, n)$. The former leads to muon decay, whereas the latter for example to beta decay, or quasi-elastic scattering $\ell\, p \to \nu_\ell\, n$. In this case, all six structures appear and setting $m_p = m_n \equiv m_N$ we have

$$\langle p(p_2)|J_{cc}^\alpha|n(p_1)\rangle = \bar{u}(p_2, m_N)\left( F_1^{(pn)}(q^2)\gamma^\alpha + F_2^{(pn)}(q^2)\frac{i\,\sigma^{\alpha\beta}q_\beta}{2\,m_N} + F_A^{(pn)}(q^2)\gamma^\alpha\gamma_5 \right. \quad (5.21)$$

$$\left. +F_P^{(pn)}(q^2)\frac{q^\alpha\gamma_5}{2\,m_N} + F_S^{(pn)}(q^2)\frac{q^\alpha}{m_N} + F_T^{(pn)}(q^2)\frac{i\,\sigma^{\alpha\beta}q_\beta\gamma_5}{2\,m_N} \right) u(p_1, m_N)\,.$$

The scalar and tensor form factors $F_S$ and $F_T$ are referred to as second-class currents and often are omitted. However, we will return to them in Section 5.6 in connection with the nucleon $\beta^-$ decay, see (5.48), which can be related to $F_{S,T}^{(pn)}$ and $F_{S,T}^{(\nu_e e^-)}$. The axial-vector and the pseudoscalar form factors, $F_A^{(pn)}$, and $F_P^{(pn)}$ are related to often used couplings as

$$g_A \equiv F_A^{(pn)}(0), \qquad \bar{g}_A \equiv F_A^{(pn)}(q_0^2), \qquad \bar{g}_P \equiv \frac{m_\mu}{m_N}F_P^{(pn)}(q_0^2), \qquad (5.22)$$

where $q_0^2 = -0.88\,m_\mu^2$ is the momentum transfer of $\mu^-$ capture on the proton, neglecting binding energies.

### 5.2.3 Hadronic effects

Not only the Wilson coefficients of the EFTs are subject to RGEs and thus scale dependent, but also the gauge couplings $\alpha = e^2/(4\pi)$ and $\alpha_s = g_s^2/(4\pi)$ in (5.1). Both depend on the energy of the phenomenon they are used to describe, but while $\alpha(Q^2)$ decreases towards $\alpha(0) \sim 1/137$, the strong coupling $\alpha_s(Q^2)$ increases as we go to lower energies. For energy scales below a couple of GeV, a perturbative expansion in $\alpha_s$ no longer works — the relevant degrees of freedom related to the strong interactions at low energies are not quarks and gluons, but light hadrons. Once more, EFTs come to the rescue, in this case chiral perturbation theory ($\chi$PT) [45–47]. As for all EFTs, the first step is to identify the relevant degrees of freedom in the energy range of interest. The second is to write down the most general Lagrangian for these degrees of freedom that is compatible with the symmetries of the underlying theory. For the strong interactions the answer to the first question is related to the phenomenon of spontaneous chiral symmetry breaking, which generates Goldstone bosons, the only massless particles of strong interactions. Actually in the spectrum of QCD there are no massless particles, but a triplet of very light pseudoscalars, the pions $\vec{\pi} = (\pi^+, \pi^0, \pi^-)$. The fact that they are not exactly massless is well understood and due to the presence of an explicit, but small, chiral symmetry breaking term in the QCD Lagrangian: the quark mass term. In the limit of zero up and down quark masses, i.e., $m_d = m_u = 0$, the three pions become massless, and since there are no other mechanisms to generate massless particles in QCD in the chiral limit, these are the only relevant degrees of freedom at low energy.

The rules to write down an effective Lagrangian for Goldstone bosons are well known. Goldstone bosons transform nonlinearly under the symmetry of the underlying theory, which leads to a non-renormalizable Lagrangian containing only derivative couplings. Symmetry constrains their interaction to become weaker as one lowers the energy. How to include an explicit symmetry breaking is also well known. The symmetry breaking parameters are promoted to spurions, fields with given transformation laws, and the effective Lagrangian must

include these fields too and still satisfy the requirement of being invariant under symmetry transformations. In the case of QCD, in addition to derivative couplings, it is also possible to have couplings proportional to the quark masses $m_{u,d}$. Clearly, there are infinitely many such terms and the Lagrangian only becomes useful with an organizing principle. Since this is a low-energy EFT, we count powers of energy or momenta as small, and since it is relativistic, they come in even powers. The smallest possible number is two, then four, six and so on. Quark masses (or explicit symmetry breaking in general) also count as small, but there is no unique choice concerning the relative importance of powers of quark masses and derivatives. The standard one is $m \sim p^2$. According to this choice the lowest-order Lagrangian contains all possible terms with two powers of derivatives or one power of quark masses and it turns out that there are only two:

$$\mathcal{L}_{\chi\mathrm{PT}} = \mathcal{L}_2 + \mathcal{L}_4 + \mathcal{L}_6 + \dots, \quad \mathcal{L}_2 = \frac{F^2}{4} \langle u_\mu u^\mu + \chi_+ \rangle, \tag{5.23}$$

where $u_\mu = iu^\dagger \partial_\mu U u^\dagger$, $\chi_+ = u^\dagger \chi u^\dagger + u \chi^\dagger u$, and

$$U = uu = \exp(i\phi/F), \quad \phi = \pi^a \tau_a, \quad \chi = 2B \operatorname{diag}(m_u, m_d), \tag{5.24}$$

with $\pi^a$ the triplet of pion fields and $\tau_a$ the Pauli matrices. The low-energy constant (LEC) $F$ is related to the pion decay constant

$$\langle 0|(J_A^a)_\mu(0)|\pi^b(p)\rangle = i\delta^{ab} F_\pi p_\mu, \quad F_\pi = F\left(1 + \mathcal{O}(m_q)\right), \tag{5.25}$$

with $(J_A^a)_\mu$ the isospin-triplet axial current. The second LEC $B$ is defined through the quark condensate in the chiral limit,

$$B = -\frac{\langle 0|\bar{u}u|0\rangle}{F^2} = -\frac{\langle 0|\bar{d}d|0\rangle}{F^2}, \tag{5.26}$$

and also relates the pion mass to the quark mass according to the Gell-Mann–Oakes–Renner relation [48]

$$m_\pi^2 = 2B\hat{m}\left(1 + \mathcal{O}(m_q)\right), \tag{5.27}$$

with $\hat{m} = (m_u + m_d)/2$. Calculating tree-level diagrams with $\mathcal{L}_2$ gives a leading-order (LO) result. Going to next-to-leading order (NLO) requires calculating one-loop diagrams with vertices only from $\mathcal{L}_2$ and tree-level diagrams with one vertex from $\mathcal{L}_4$ [32, 46]. At next-to-next-to leading order (NNLO) two-loop diagrams with vertices only from $\mathcal{L}_2$, one-loop diagrams with one vertex from $\mathcal{L}_4$ and tree-level diagrams with two vertices from $\mathcal{L}_4$ or one from $\mathcal{L}_6$ contribute [49–51], and so on.

The limit of validity of this EFT is given by the scale of chiral symmetry breaking. In the expansion in powers of momenta and quark masses that is generated by the effective Lagrangian above, the relevant scale is represented by $\Lambda_\chi = 4\pi F_\pi \sim 1.2$ GeV. Physically it represents the scale at which degrees of freedom other than Goldstone bosons get excited, such as the $\rho$, whose mass $m_\rho \sim 0.77$ GeV is indeed close to $\Lambda_\chi$.

The same approach also works for other particles beyond the pions. In the limit $m_s \to 0$ also the kaons and the eta become Goldstone bosons and can be included in the formalism above [52]. The field $\phi$ becomes a $3 \times 3$ matrix containing the octet of Goldstone bosons $\phi = \phi^a \lambda_a$, and $\chi$ has to be trivially extended to a diagonal $3 \times 3$ quark-mass matrix.

A less trivial extension concerns the baryon sector [53–56]. At first sight this would seem impossible, since the mass of the nucleons is close to $\Lambda_\chi$. But the baryon number $n_B$ is conserved in strong interactions and one can split the spectrum in separated sectors, labeled by

$n_B$. Quantities like the nucleon masses, their form factors, or their scattering amplitude with a pion (or any other Goldstone boson(s)) all belong to the sector $n_B = 1$ and can also be studied with the help of the chiral expansion. In this case this represents an expansion in powers of momenta and quark masses around the ground-state energy, which in this sector is equal to the mass of the nucleon $m_N$, rather than zero.

From the point of view of their transformation properties, nucleons are spin-1/2 as well as isospin-1/2 particles, and transform linearly under chiral transformations. In particular the fact that they are spin-1/2 particles has an important consequence as the expansion of the Lagrangian in powers of momenta (derivatives) contains both even and odd powers

$$\mathcal{L}_N = \mathcal{L}_1 + \mathcal{L}_2 + \mathcal{L}_3 + \dots \tag{5.28}$$

The leading-order Lagrangian looks as follows

$$\mathcal{L}_1 = \bar{N}(i\slashed{D} - m)N + \frac{1}{2} g_A \bar{N} \slashed{u} \gamma_5 N, \tag{5.29}$$

with the covariant derivative defined as

$$D_\mu = \partial_\mu + \Gamma_\mu, \quad \Gamma_\mu = \frac{1}{2}[u^\dagger, \partial_\mu u], \tag{5.30}$$

and $\bar{N} = (\bar{p}, \bar{n})$ the isospin doublet containing the Dirac spinors of the proton and neutron. The parameters $m$ and $g_A$ represent the mass and the axial coupling of the nucleon in the chiral limit, respectively. Note that the chiral symmetry imposes the presence of the pion field both in the covariant derivative as well as in the coupling to the nucleon axial current. From this follows the famous Golberger-Treiman relation [57]

$$g_{\pi N} = \frac{g_A m_N}{F_\pi}, \tag{5.31}$$

between the pion-nucleon coupling constant $g_{\pi N}$ (whose square is the residue of the nucleon pole in the $\pi N$ scattering amplitude), the physical nucleon mass, and the axial coupling.

The low-energy description of the strong-interaction effects in terms of $\chi$PT cannot only be formulated for pure QCD as the underlying theory. While QED effects can be included in terms of explicit low-energy degrees of freedom, the chiral realization of higher-dimensional operators again is based on the external-field and spurion technique. Traditionally, this has been done to include weak-interaction effects and it can be generalized to include BSM effects encoded in the LEFT Lagrangian (5.16).

## 5.3 The muon

The muon is a fundamental lepton similar to the electron, however with a much larger mass, $m_\mu \simeq 105.66\,\text{MeV}$. It is unstable and predominantly decays through the Michel process

$$\mu \to e \nu \bar{\nu}, \tag{5.32}$$

which leads[2] to a lifetime of about $\tau_\mu \simeq 2.2\,\mu\text{s}$. As discussed in the context of (5.21) the decay is mediated by the charged current $J_{cc}^\alpha$, leading to a non-vanishing current-current interaction $\langle \nu_\mu | J_{cc}^\alpha | \mu \rangle \langle e | (J_{cc})_\alpha^\dagger | \nu_e \rangle$. From an EFT point of view this corresponds to a four-fermion operator $(\bar{\nu}_\mu \gamma^\alpha P_L \mu)(\bar{e} \gamma_\alpha P_L \nu_e)$ and its Hermitian conjugate. For computational reasons it is more

---

[2] Section 16: MuLan [2].

convenient to work with the Fierz transform of this operator. This results in the Fermi theory, an EFT defined through the Lagrangian

$$\mathcal{L}_{\text{Fermi}} = -\frac{4\,G_F}{\sqrt{2}} \left( \bar{\nu}_\mu \gamma_\alpha P_L \nu_e \right) \left( \bar{e} \gamma^\alpha P_L \mu \right) + \text{h.c.} + \mathcal{L}_{\text{QED+QCD}}, \tag{5.33}$$

where it is implicitly assumed that only light quarks are included in $\mathcal{L}_{\text{QCD}}$. The first term on the r.h.s. of (5.33) corresponds to the operator $[O_{\nu\ell}^{V,LL}]_{2112}$ as introduced in (5.12). Its Wilson coefficient, $4\,G_F/\sqrt{2}$, has the special property that it does not get renormalized [58]. Thus, the Lagrangian (5.33) can be used to consistently compute at leading order in $G_F$ but to all orders in the electromagnetic coupling $\alpha$. Only the usual QED renormalization procedure has to be applied. As an example, the lifetime of the muon can be expressed as

$$\frac{1}{\tau_\mu} \equiv \Gamma_\mu = \Gamma_0 \left( 1 + \Delta q \right) = \frac{G_F^2 m_\mu^5}{192\,\pi^3} \left( 1 + \Delta q \right), \tag{5.34}$$

where $\Delta q$ contains all corrections to $\Gamma_0$ (the tree-level result for massless electrons) that are induced by (5.33). This includes electron-mass effects, higher-order QED corrections, as well as hadronic corrections. While the former two can be computed in perturbation theory, the latter are more delicate. As mentioned above, QCD is non-perturbative at scales typical for muonic processes, $q^2 \sim m_\mu^2$. Thus, the hadronic contributions have to be determined by other means. This is often the leading theoretical uncertainty. The fact that such corrections for muonic processes enter only at NNLO makes the muon a rather clean laboratory for precision physics. Typically, $\mathcal{L}_{\text{QED}}$ contains muon and electron fields, but the inclusion of $\tau$ leptons is straightforward, as is the inclusion of heavy quarks in $\mathcal{L}_{\text{QCD}}$.

The corrections $\Delta q$ are known at NNLO with full electron mass dependence [59–62]. Thus, with a precision measurement of the muon lifetime, the Wilson coefficient in (5.33), or equivalently $G_F$, can be determined extremely precisely. This, in turn, is an important input for electroweak precision tests. In fact, $G_F$ can be related to $m_W$ and $m_Z$ through

$$\frac{4\,G_F}{\sqrt{2}} = \frac{g^2}{2m_W^2} \left( 1 + \Delta r \right) = \frac{2\pi\,\alpha}{\sin^2\theta_W\, m_W^2} \left( 1 + \Delta r \right), \tag{5.35}$$

where (in the on-shell scheme) $\sin^2\theta_W = 1 - m_W^2/m_Z^2$. The SM corrections $\Delta r$ contain (partially hadronic) fermion loop contributions to the charge renormalization. Additional contributions depend also on the top and Higgs mass. This makes $G_F$ a decisive input for SM consistency checks. As mentioned in [2] only the availability of the NNLO result [59] allowed for a full exploitation of the experimental results.

While SM corrections are crucial for the electroweak precision tests the tree-level matching of the SM to the Fermi theory yields the matching condition (5.35) with $\Delta r \to 0$ that is used in (5.33). Furthermore, terms of order $q^2/m_W^2$ relative to the four-fermion interaction are also neglected in (5.33) and typically in (5.16). In the literature (5.34) is often written with an additional factor $(1 + 3/5\,(m_\mu/m_W)^2)$ which results in a $10^{-6}$ correction. Within the EFT, such corrections are reproduced by dimension-8 operators, which are missing in (5.33). There are also numerous dimension-6 operators generated by the SM that are not included in (5.33). The corresponding Wilson coefficients are related to the general parametrization of muon decay parameters.[1]

Apart from the Michel decay, two further SM decay processes are of interest; the radiative and rare decays

$$\mu \to e\,\nu\bar{\nu}\gamma, \qquad\qquad \mu \to e\,\nu\bar{\nu}e^+e^-. \tag{5.36}$$

In order to be well defined and to avoid infrared singularities, the branching ratio for the radiative decay must be defined requiring a minimal energy of the photon. For $E_\gamma > 10$ MeV we have $B(\mu \to e\nu\bar{\nu}\gamma) \sim 1.3 \times 10^{-2}$. For the rare decay the branching ratio is $B(\mu \to e\nu\bar{\nu}ee) \sim 3.6 \times 10^{-5}$. A fully differential NLO description of these processes in the Fermi theory (5.33) is available [63–66]. Depending on the cuts that are applied, the NLO QED corrections can be sizeable. Experimental information on the branching ratio of the radiative decay has been obtained by MEG [67] and PiBeta [68].

A particularly attractive feature of particle physics with muons is the study of cLFV decays. There are three "golden" channels

$$\mu \to e\gamma\,, \qquad\qquad \mu \to eee\,, \qquad\qquad \mu^- {}^A_Z N \to e^- {}^A_Z N\,. \qquad (5.37)$$

PSI has a long tradition in corresponding experimental searches.[3,4,5,6] For the first two processes typically $\mu^+$ are used, whereas $\mu^-$ must be used for muon conversion in the field of a nucleus ${}^A_Z N$ with atomic number Z and mass number A. In the SM (with non-vanishing neutrino masses) the branching ratios for these processes are smaller than $10^{-50}$, but not zero [69]. Hence, from a theory point of view there is nothing sacred about lepton flavor. As we know that it is not conserved, it is very natural to expect much larger cLFV branching ratios in BSM than in the SM. In fact, generic extensions of the SM do typically lead to large cLFV rates and suppressing them requires additional tuning or model-building efforts.

To extract constraints on BSM physics from limits on the branching ratios of the processes (5.37), they are computed in $\mathcal{L}_{\text{LEFT}}$, typically at tree level. For $\mu \to e\gamma$ the dipole operator $[O^D_{\ell\gamma}]_{21}$ (5.14) enters. Thus we get a limit on the corresponding Wilson coefficient at the low scale $[C^D_{\ell\gamma}]_{21}(m_\mu)$. In a next step, the RGE is used to convert this to limits for the Wilson coefficients at the high scale, $C_i(\Lambda)$. Some scalar four-fermion interactions mix at NLO whereas vector four-fermion interactions enter at NNLO. Nevertheless, this results in very stringent limits on contact interactions induced by BSM physics. They have to be combined with limits from $\mu \to eee$ and muon conversion, where contact interactions already appear at leading order. Using as many operators as possible in connection with RGE maximizes the information that can be obtained from low-energy observables.

These computations can be made [70] for $\mu \to e\gamma$ and $\mu \to eee$ using standard perturbative methods with the Lagrangian (5.16), although for some contributions, non-perturbative effects play a role [71]. However, additional input is required for muon conversion. First, the nuclear matrix elements $\langle {}^A_Z N | J | {}^A_Z N \rangle$ for vector and scalar currents/operators are required. The former can be obtained trivially through current conversion, but the latter need input from lattice QCD or $\chi$PT. Second, the overlap integrals of the lepton wave function with the nucleus are required [72]. In principle different target nuclei provide different limits on the various coefficients, but in practice the model discriminating power is limited [73]. A further complication is due to background from the decay in orbit (DIO). This is the Michel decay of the $\mu^-$ bound in the nucleus

$$\mu^- {}^A_Z N \to e^- \nu_\mu \bar{\nu}_e {}^A_Z N\,. \qquad (5.38)$$

Due to nuclear recoil effects the energy spectrum of the electron has a tail up to $m_\mu$, the energy of the signal for the electron from muon conversion. Thus DIO has to be studied as a background process [74].

So far the nucleus has acted only as a spectator. The only nuclear physics that was required is the nuclear matrix element. For completeness we mention here two processes relevant to

---

[3] Section 7: SINDRUM [3].

[4] Section 8: SINDRUM II [4].

[5] Section 19: MEG [5].

[6] Section 20: Mu3e [6].

muon conversion, where the nuclear physics is much more involved. When the $\mu^-$ is bound to the nucleus, it quickly cascades to the $1S$ ground state. Then it might undergo muon capture

$$\mu^- \, {}^A_Z N \to \nu_\mu \, {}^A_{Z-1} N \tag{5.39}$$

before it decays. The corresponding nuclear matrix element $\langle {}^A_{Z-1} N | (J^\alpha_{cc})^\dagger | {}^A_Z N \rangle$ is an extended version of (5.21). It depends on the details of ${}^A_Z N$ and is not easily accessible with theoretical methods. We will return to muon capture in Section 5.4.

The muon can not only form bound states with a nucleus, but also with an electron. Muonium, $M = (\mu^+ e^-)$, is a bound state like hydrogen, but with the proton replaced by a positive muon. As the latter is a pointlike fermion, muonium is an excellent laboratory for QED tests, and for a precise determination of the muon mass.[7] As the muonium mass is dominated by antimatter, $M$ is also an interesting option to study experimentally gravity of antimatter [75]. In addition, muonium-antimuonium oscillations

$$M = (\mu^+ e^-) \leftrightarrow \bar{M} = (\mu^- e^+), \tag{5.40}$$

which are forbidden in the SM, are another channel to scrutinize BSM physics.[8] A bound state of two muons, true muonium ($\mu^+ \mu^-$), is unfortunately, not experimentally accessible in the foreseeable future.

Two further properties of the muon that are of utmost importance are the AMM (5.4) and EDM (5.19). The motivation to study them in detail is again driven by the desire to test the SM. For the AMM very precise measurements are confronted with similarly precise theoretical predictions [76]. At the time of writing, there is an intriguing tension between SM theory and experiment. For the EDM, the situation is similar to cLFV searches in that the SM value is zero for practical experimental purposes. Hence, experimental verification of a non-vanishing muon EDM is a clear indication of BSM. So far, these quantities have not been measured by PSI experiments. However, future involvement, in particular for the EDM, is being considered [77].

## 5.4 The proton

Like the electron and muon, the proton is a charged spin $1/2$ fermion. However, because the proton is a bound state, the form factors (5.5) cannot be computed perturbatively simply using $\mathcal{L}_{QED+QCD}$. Most information is obtained from experiment, with additional input from lattice QCD and $\chi$PT [78]. From the charge and measurements of the AMM we know $F_1^{(p)}(0) = 1$ and $F_2^{(p)}(0) = \kappa_p \simeq 1.79$.

A quantity that has received a lot of attention in the past years is the proton charge radius $r_E^{(p)}$. As discussed in the context of (5.8), the radius can be extracted as the slope of $G_E^{(p)}(q^2)$ at $q^2 \to 0$. This can be determined by low-$q^2$ lepton-proton scattering with a careful $q^2 \to 0$ extrapolation. An alternative approach is to use spectroscopy of normal hydrogen or better muonic hydrogen. The overlap of the lepton wave function with the proton charge distribution impacts on the energy levels. Thus, a precise measurement of different transition energies allows the extraction of information on the proton radius. As the Bohr radius is proportional to $1/m_\ell$, the effect in muonic atoms is considerably larger. This has resulted in a very precise new determination of the proton radius[9] and a new world average of $r_E^{(p)} \simeq 0.84$ fm. The disagreement with earlier determinations of $r_E^{(p)}$ was referred to as proton radius puzzle [79,80], but the puzzle is fading away [81].

---

[7] Section 29: MSpec, Mu-Mass [7].
[8] Section 9: MACS [8].
[9] Section 21: CREMA [9].

The CREMA collaboration[9] has measured two transition frequencies for muonic hydrogen; the triplet $E\left(2P_{3/2}^{F=2}\right) - E\left(2S_{1/2}^{F=1}\right)$ and singlet $E\left(2P_{3/2}^{F=1}\right) - E\left(2S_{1/2}^{F=0}\right)$. From these two values and theoretical input for the fine structure, it is possible to extract the Lamb shift $E_L = E\left(2P_{1/2}\right) - E\left(2S_{1/2}\right)$ and the hyperfine splitting $E_{HFS} = E\left(2S_{1/2}^{F=1}\right) - E\left(2S_{1/2}^{F=0}\right)$. The discrepancy of the proton radius determination from muonic hydrogen with earlier values initiated a flurry of activities to revisit the theoretical calculations of the energy levels, as summarized in [82]. This involves radiative corrections and recoil effects, which can in principle be computed in perturbation theory.

In addition there are proton-structure effects, which are divided into two categories: a) finite-size effects, which depend on the charge $\rho_E$ and magnetic moment distribution $\rho_M$ of the proton, i.e., the charges related to the form factors $G_E^{(p)}$ and $G_M^{(p)}$, introduced in (5.6); b) polarizability effects.

The leading finite-size effect for $E_L$ is in fact proportional to $\left(r_E^{(p)}\right)^2$ and it is precisely this effect that allows an accurate determination of $r_E^{(p)}$ from muonic hydrogen spectroscopy to be made. There are also higher-order effects which have to be included, most notably a contribution from the so-called third Zemach moment

$$\left(r_F^{(p)}\right)^3 \equiv \frac{48}{\pi} \int_0^\infty \frac{dQ}{Q^4} \left( \left[G_E^{(p)}(Q^2)\right]^2 - 1 + \frac{1}{3}\left[r_E^{(p)}\right]^2 Q^2 \right), \tag{5.41}$$

where $r_F^{(p)}$ is referred to as Friar radius. This contribution is related to the elastic two-photon exchange (TPE), where elastic refers to the fact that the intermediate hadronic state is still a proton. The inelastic TPE, i.e., TPE where the intermediate hadronic state is more complicated, is often referred to as polarizability correction.

A similar distinction between perturbative and finite-size contributions can be made for the hyperfine splitting $E_{HFS}$. In this case, the leading finite-size effect is proportional to the Zemach radius $r_Z^{(p)} \simeq 1.0$ fm, a convolution of the charge distribution with the magnetic moment distribution

$$r_Z^{(p)} \equiv \int d^3\vec{r}_1 \int d^3\vec{r}_2 \, \rho_E^{(p)}(\vec{r}_1)\rho_M^{(p)}(\vec{r}_2)|\vec{r}_1 - \vec{r}_2|. \tag{5.42}$$

While the determination of the magnetic radius of the proton $r_M^{(p)} \simeq 0.8$ fm was discussed less controversially, there is also quite a spread in the values obtained from different extractions [83]. This spread is typically attributed to different treatment of TPE contributions.

The CREMA collaboration also investigated muonic deuterium and helium[9] and determined the corresponding charge radii. Measuring the charge radii of higher $Z$ nuclei[10] provides crucial input for potential atomic parity violation experiments.

Returning to the proton, as mentioned above, studying lepton-proton scattering at low $q^2$ is an important source to obtain information on the proton form factors and, hence, the proton radius. At tree level, which implies the one-photon approximation, this process is described by the famous Rosenbluth formula

$$\frac{d\sigma}{d\Omega} = \frac{\alpha^2}{4E_1^2\sin^4\theta_2}\frac{E_3}{E_1}\left(\frac{\left[G_E^{(p)}(q^2)\right]^2 + \tau\left[G_M^{(p)}(q^2)\right]^2}{1+\tau}\cos^2\theta_2 + 2\tau\left[G_M^{(p)}(q^2)\right]^2\sin^2\theta_2\right), \tag{5.43}$$

---

[10] Section 22: muX [10].

in terms of $\tau = -q^2/(4m_p^2)$, the scattering angle $\theta = 2\theta_2$, and the energies of the incoming and outgoing leptons, $E_1$ and $E_3$, respectively. Using the standard dipole form $G_D(q^2)$ for the form factors gives a good fit to the experimental data:

$$G_E^{(p)}(q^2) \simeq \frac{G_M^{(p)}(q^2)}{1 + \kappa_p} \simeq G_D(q^2) = \frac{1}{(1 - q^2/\Lambda^2)^2} \qquad \text{with} \quad \Lambda^2 = 0.71 \, \text{GeV}^2. \tag{5.44}$$

For very small $q^2$ the form factors deviate from (5.44) and — coming back to the proton radius issue — it is a delicate problem to extract the slope of the form factors in the limit $q^2 \to 0$ from scattering data.

Given the importance of lepton-proton scattering, there is a vast literature on the computation of higher-order corrections to (5.43). These corrections can be split into gauge independent and finite subsets by separately considering radiative corrections from the lepton line, radiation from the proton line, and multi-photon exchange between the proton and electron.

A full NLO calculation, superseding earlier ones where various approximations had been used, has been presented in [84] and there are several Monte Carlo generators with these corrections implemented [85, 86]. Corrections at NNLO due to radiation from the electron line have also been computed [87, 88]. Due to the small mass of the lepton, these are the dominant corrections, particularly for electron-proton scattering. As for spectroscopy, from a theoretical point of view, multi-photon exchange contributions between the lepton and proton are the most difficult ones to handle. Accordingly, TPE contributions have received a lot of attention, also including the inelastic parts, see e.g. [89–92].

Traditionally, these experiments have been carried out with electrons. The MUSE collaboration[11] proposes to measure $\ell p \to \ell p$ with $\ell \in \{e^\pm, \mu^\pm\}$. This offers the opportunity to compare $e p$ and $\mu p$ scattering within the same experimental setup. In addition, experimental information on TPE can be obtained by measuring the difference between $\ell^+ p$ and $\ell^- p$ scattering.

To the best of our knowledge, the proton is a stable particle and in all processes discussed so far, has been left intact. A low-energy process that affects the proton much more dramatically is muon capture, $\mu^- p \to n \nu_\mu$. This process can be described by the transition matrix element (5.21) as a current-current interaction $\langle \nu_\mu | J_{cc}^\alpha | \mu \rangle \langle n | (J_{cc})_\alpha^\dagger | p \rangle$. In fact, muon capture on the proton as measured by MuCap[12] gives valuable information on the corresponding form factors, in particular $\bar{g}_P$ (5.22) [93]. The inverse process would be related to neutrino-nucleon scattering. Muon capture on the deuterium has been investigated by MuSun.[13]

## 5.5  Nucleons and nuclei

The proton and neutron together form an isospin doublet. They differ by their isospin projection, $I_3 = +1/2$ and $I_3 = -1/2$, and quark content, $uud$ and $udd$, respectively. The neutron's Dirac and Pauli form factors are normalized as $F_1^{(n)}(0) = 0$ and $F_2^{(n)}(0) = \kappa_n \simeq -1.91$. The former differs from the proton form factor at zero momentum transfer, $F_1^{(p)}(0) = 1$, due to the vanishing charge of the neutron. Therefore, the electric Sachs form factor of the neutron cannot be approximated with a dipole form factor (5.44). Instead, the Galster form factor could be used as a simple parametrization [94]:

$$G_E^{(n)}(q^2) = \frac{q^2 \kappa_n}{4m_n^2 - \eta q^2} G_D(q^2), \tag{5.45}$$

---

[11] Section 23: MUSE [11].
[12] Section 17: MuCap [12].
[13] Section 18: MuSun [13].

with $\eta = 5.6$. Since there are no free neutron targets, one has to rely on scattering off light nuclei (e.g., $^2$H or $^3$He) to extract the neutron form factors and polarizabilities. Thereby, few-nucleon EFTs are needed to separate the neutron from proton and nuclear effects.

As highlighted in the previous section, muonic atoms are sensitive to the nuclear structure. The measurement of the muonic-hydrogen Lamb shift by the CREMA collaboration[9] allowed the extraction of the proton root-mean-square charge radius with unprecedented precision. From the measured the Lamb shifts in $\mu$D, $\mu^3$He$^+$ and $\mu^4$He$^+$ the deuteron, helion and $\alpha$-particle charge radii can be extracted. In the future, the ground-state hyperfine splitting of $\mu^3$He$^+$ shall be measured to extract the helion Zemach radius. To extract the different nuclear radii, precise theory predictions for the energy levels in muonic atoms are needed, see theory summaries in [95–97]. Among other contributions, one needs the finite-size effects, through which the different radii enter, and the polarizability effects. For the light muonic atoms, not only the proton polarizability enters, but also the polarizabilities of the neutron and the nucleus as a whole. Similar complications arise when going from pionic hydrogen to pionic deuterium[14] or helium.[15] The nuclear polarizabilities are typically several orders of magnitude larger than the nucleon polarizabilities, and thus, more important. Take for instance the electric dipole polarizability, $\alpha_{E1}^{(n)} = 11.8(1.1) \times 10^{-4}$ fm$^3$ [98] and $\alpha_{E1}^{(d)} = 0.6314(19)$ fm$^3$ [99], which describes the deformation of a composite particle in an external electric field and gives a dominant contribution to the two-photon exchange. The nuclear polarizability effects can be calculated in a dispersion relations framework [100, 101] or based on nuclear potentials. For the latter, one distinguishes calculations with phenomenological models [102] fit to nucleon-nucleon scattering data, such as the AV18 potential [103], or with nucleon-nucleon interactions derived from chiral EFT [104–107]. The nucleon-structure contributions are often deduced by rescaling the proton-structure contributions to $\mu$H. Take, for example, the nucleon-polarizability contribution

$$\delta_{\mathrm{pol}}^{\mathrm{N}}(\mu A) = (N + Z)\left[Z m_r(\mu A)/m_r(\mu H)\right]^3 \delta_{\mathrm{pol}}^{\mathrm{N}}(\mu H), \qquad (5.46)$$

where $m_r$ is the reduced mass of the muonic atom and Z, N, A are the numbers of protons, neutrons and nucleons in the nucleus.

Also in the field of muonic atoms, the muX project[10] determines nuclear charge radii of radioactive elements and rare isotopes, e.g, $^{248}$Cm and $^{226}$Ra, through muonic X-ray measurements. These are needed as input for atomic parity violation experiments. In addition, muX probes nuclei that are at the end of a double $\beta$ decay chain. These are interesting in view of possible neutrinoless double $\beta$ decay that could occur if neutrinos were Majorana particles. Two examples are the following $\beta^-\beta^-$ decays:

$$^{130}_{52}\text{Te} \xrightarrow{\beta^-} {}^{130}_{53}\text{I} \xrightarrow{\beta^-} {}^{130}_{54}\text{Xe},$$

$$^{82}_{34}\text{Se} \xrightarrow{\beta^-} {}^{82}_{35}\text{Br} \xrightarrow{\beta^-} {}^{82}_{36}\text{Kr}.$$

Here one uses muon capture to study excited states of $^{130}$Xe and $^{82}$Kr. In the future, direct searches for BSM interactions between muons and nuclei might be possible with the muX setup.

To further advance the precision of the few nucleon EFTs mentioned in this section, the MuSun experiment[13] is studying muon-capture on deuterium: $\mu^- d \to nn\nu_\mu$. The aim is to determine the LEC of the axial-vector four-nucleon interaction $d$ [108]

$$\mathcal{L}_{NN} = -2d(N^\dagger S \cdot u N)N^\dagger N, \qquad (5.47)$$

---

[14] Section 14: Pionic hydrogen and deuterium [14].
[15] Section 26: Pionic helium [15].

where $S^\mu$ is the nucleon covariant spin operator, $N(x)$ is the nucleon field, and $u_\mu$ is given below (5.23). Presently, this LEC has only been extracted from $A = 3$ nuclei. The MuSun experiment has the potential for an improved extraction at the 20 % level.

## 5.6 The free neutron

In the previous section, we discussed nuclei and bound neutrons. In the following, we discuss free neutrons provided by the Swiss Spallation Neutron Source (SINQ) and the PSI Ultra Cold Neutron (UCN) source [109]. As we will see, the neutron experiments at PSI are dedicated to BSM searches, and in particular, to the search for CP violation in the light quark sector.

The neutron is unstable with a lifetime of about 880 s. The long-standing tension between measurements with in-flight and stored neutrons has led to speculations that there could be 'dark' BSM decay channels [110, 111]. Within the SM, the neutron decays into the proton, where the dominant decay channel is the classical $\beta^-$ decay $n \to pe^-\bar{\nu}_e$, described by the current-current interaction from the Fermi theory, (5.11). Besides the dominant $V-A$ structure of the weak interaction, there could be small admixtures of scalar and tensor couplings. Using the general formulation of Lee and Yang, which is an older version of the parametrization in (5.21), the $\beta^-$ decay reads [112]

$$
\begin{aligned}
\langle pe^-\bar{\nu}_e|n\rangle = \frac{G_F V_{ud}}{\sqrt{2}}\Bigg[ & \langle p|n\rangle\langle e^-|C_S - C_S'\gamma_5|\nu_e\rangle + \langle p|\gamma_\mu|n\rangle\langle e^-|\gamma^\mu\left(C_V - C_V'\gamma_5\right)|\nu_e\rangle \\
& + \tfrac{1}{2}\langle p|\sigma_{\lambda\mu}|n\rangle\langle e^-|\sigma^{\lambda\mu}\left(C_T - C_T'\gamma_5\right)|\nu_e\rangle - \langle p|\gamma_\mu\gamma_5|n\rangle\langle e^-|\gamma^\mu\gamma_5\left(C_A - C_A'\gamma_5\right)|\nu_e\rangle \\
& + \langle p|\gamma_5|n\rangle\langle e^-|\gamma_5\left(C_P - C_P'\gamma_5\right)|\nu_e\rangle + \text{h.c.}\Bigg],
\end{aligned}
\tag{5.48}
$$

where $C_i^{(\prime)}$ are 10 complex coupling constants. For the SM with conserved vector current, $g_V = 1$, the only non-vanishing couplings are $C_V = C_V' = 1$ and $C_A = C_A' = -g_A$. Parity violation is assured if $C_i \neq 0$ and $C_i' \neq 0$. Time reversal violation (TRV), or CP violation, is found if $\text{Im}(C_i/C_j) \neq 0$ or $\text{Im}(C_i'/C_j) \neq 0$, i.e., if at least one coupling has an imaginary phase relative to the others. The nTRV experiment[16] accessed the scalar and tensor couplings through the measurement of the transverse polarization of electrons from the decay of polarized free neutrons. At the present level of precision, the results are in agreement with the SM, thus, setting constraints on BSM physics. For a review on electroweak SM tests with nuclear $\beta$ decays see [113].

The observation of a nonzero permanent EDM of the neutron could be interpreted as a signal of CP violating BSM interactions or a measurement of the QCD $\theta$ parameter, see (5.20). The current best limit $|d_n| < 1.8 \times 10^{-26}\,e\,\text{cm}$ is from the nEDM experiment[17] at PSI. This limit is still compatible with the CKM-induced SM contributions to $d_n$, which are negligible as explained below (5.19). The n2EDM experiment will improve the sensitivity to $d_n$ by an order of magnitude and probe BSM physics at the multi-TeV scale [43]. The electric field of these experiments is of the order of $10^6\,\text{V/m}$. This is well below the critical electric field strength, $E_{\text{crit.}} \sim 10^{23}\,\text{V/m}$, that would be able to induce an EDM proportional to the neutron electric dipole polarizability $d_{\text{ind.}} = 4\pi\alpha_{E1}\vec{E}$ [114]. The nEDM spectrometer has also been used in indirect searches for Dark Matter (DM) candidates, e.g., mirror matter or axions and axion-like particles (ALPs).[18]

---

[16] Section 15: nTRV [16].
[17] Section 27: nEDM [17].
[18] Section 28: nEDMX [18].

## 5.7 The pion

Low-energy pion physics provides access to a large variety of phenomena, ranging from strong non-perturbative dynamics over electroweak precision tests to probes of BSM physics. The pions are stable in pure QCD and as asymptotic QCD states they play a special role in many hadronic processes, where they appear as hadronic final states. Pion interactions can be understood beyond the chiral expansion by employing unitarity and analyticity of transition amplitudes, which provide a means to resum pion-rescattering effects. Most notably, $\pi\pi$ scattering has been accurately described in terms of the Roy equations [115–117], and the resulting precise determination of the scattering phase shifts provides a central input in the analysis of a host of other hadronic processes at low energies.

An important probe of QCD at low energies is provided by the interaction of pions with nucleons. Pionic atoms provide access to $S$-wave $\pi N$ scattering lengths [118], because the strong interaction changes the spectrum compared to pure QED, resulting in shifts of the energy levels and in finite widths of the bound state. The most precise measurements of pionic hydrogen and deuterium have been performed at PSI.[14] The $S$-wave scattering lengths enter as important constraints in a dispersive Roy–Steiner analysis of the $\pi N$ scattering amplitude [119].

Compared to pure strong dynamics in the isospin limit, both electromagnetic effects and the mass difference between up and down quarks generate small isospin-breaking corrections. The mass difference of charged and neutral pions is understood to arise almost exclusively from electromagnetic effects [46, 120, 121]. This mass difference $m_{\pi^-} - m_{\pi^0}$ has been determined with high precision at PSI[19] starting from $(\pi^- p)$ bound states with subsequent charge-exchange reaction $\pi^- p \to \pi^0 n$. $m_{\pi^-}$ has also been determined at PSI by measuring the energy spectrum of pionic hydrogen $(\pi^- p)$.[20]

In the presence of electromagnetism, the neutral pion is not a stable particle, and decays predominantly into two photons. The decay results from the anomalous non-conservation of the axial current that couples to the pion. Quark-mass and electromagnetic corrections to the leading Adler–Bell–Jackiw anomaly have been worked out [122, 123]. Further decay modes, such as $\pi^0 \to e^+ e^- \gamma$, $\pi^0 \to 4e$, and $\pi^0 \to e^+ e^-$ involve the transition $\pi^0 \to \gamma^* \gamma^{(*)}$ with one or two virtual photons. The transition form factor for this process has received considerable interest in connection with hadronic contributions to the muon anomalous magnetic moment [76, 124–126].

Charged pions only decay due to the weak interaction. The hadronic part of the decay rate for $\pi^+ \to \ell^+ \nu_\ell$ is governed by the pion decay constant $F_\pi$ of (5.25), whereas the leptonic part results in a helicity suppression by a factor $m_\ell^2$. Hence, the muonic decay mode dominates over the electronic mode and has been used to measure[21] the mass of $\pi^+$. Several other decay modes have been measured at PSI by the SINDRUM,[3] PiBeta,[22] and PEN[23] experiments, including the radiative decays $\pi^+ \to \ell^+ \nu_\ell \gamma$ and $\pi^+ \to e^+ \nu_e e^+ e^-$ and pion beta decay[22] $\pi^+ \to \pi^0 e^+ \nu_e$. The theoretical description of the radiative decay $\pi^+ \to \ell^+ \nu_\ell \gamma$ is split into two parts, the so-called inner bremsstrahlung contributions (IB) and the structure-dependent terms (SD). The IB consist of the normal pion decay with additional emission of a photon from the charged external legs. This part depends on $F_\pi$. The SD terms require a more involved parametrization of the QCD effects in terms of two form factors. Apart from an axial form factor $F_A$ also a vector form factor $F_V$ contributes [127].

The charged-pion decays probe the weak interaction in the low-energy regime, where an excellent description is provided by Fermi's effective theory of current-current interaction, or

---

[19] Section 12: neutral pions [21].
[20] Section 10: negative pions [19].
[21] Section 11: positive pions [20].
[22] Section 24: PiBeta [22].
[23] Section 25: PEN [23].

more generally the LEFT framework explained in Section 5.2. The relevant operator is

$$\mathcal{L}_{\text{LEFT}} \supset \sum_{i,j,k,l} C^{V,LL}_{\underset{ijkl}{vedu}} (\bar{\nu}_i \gamma^\alpha P_L \ell_j)(\bar{d}_k \gamma_\alpha P_L u_l) + \text{h.c.,} \tag{5.49}$$

with flavor indices $i, j, k, l$ and the SM tree-level matching at the weak scale given by $C^{V,LL}_{\underset{ijkl}{vedu}} = -\frac{4G_F}{\sqrt{2}}\delta_{ij}V^\dagger_{kl}$. Therefore, the pion decays probe the CKM matrix element $V_{ud}$, with a value of $|V_{ud}| = 0.9739(27)$ resulting from the PiBeta measurement of pion beta decay. Although precise, this value is not competitive with determinations from superallowed nuclear beta decays [98], which currently are in some tension with first-row CKM unitarity. With the absence of nuclear structure aspects and with radiative corrections under good theoretical control [128], pion beta decays are theoretically clean but remain experimentally challenging due to the tiny branching ratio $\sim 10^{-8}$.

Additional semileptonic operators in the LEFT Lagrangian with different Dirac structures parametrize deviations from the SM and can be probed by several pion decay modes [129]. E.g., strong constraints on the first-generation tensor-operator coefficient $\text{Re}(C^{T,RR}_{vedu})$ arise from the $\pi^+ \to e^+ \nu_e \gamma$ Dalitz-plot study of the PiBeta experiment.

## 5.8 Conclusions

Low-energy, high-precision experiments provide essential input to improve our understanding of the fundamental interactions. They complement and extend information obtained from the energy frontier. EFTs are the theoretical tool of choice to describe and interpret their results and indeed they are well suited to describe both the SM and potential deviations therefrom in a model-independent way. In particular it is possible, and crucial, to analyze if potential deviations from the SM in different observables are linked and have a common explanation. There are numerous examples where low-energy constraints rule out apparently attractive new physics scenarios. A broad and vigorous world-wide low-energy experimental program is indispensable to make further progress in testing the SM and searching for physics beyond. Past and future experiments at PSI will continue to play their part in this challenge.

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
