# Peer review of "A theory vade mecum for PSI experiments"

_SciPost Physics Proceedings, doi:SciPost Phys. Proc. 5, 005 (2021)_

## Round 1 · Referee Report · Anonymous (Referee 1) · 2021-6-28

Report
The paper is not (fully) suitable for publication in SciPost in its present
form. The authors provide a comprehensive theoretical introduction to PSI
experiments. However, there are a few (minor) issues that require some
modifications.
(i) I see that Refs. [1-23] are not final. I am expecting these citations to
be updated before this contribution will be published.
(ii) Related to Eq. (5.12) it might be worth to refer to the LE-Fermi limit of
the SM, i.e. that the Wilson coefficients of these operators may even acquire
higher-order corrections if matched to a particular model (as the SM). I am
emphasizing this very particular issue, since these matching conditions gave a
lot of information about the top mass and Higgs mass (after knowing the top
mass). I think that the SM in the low-energy limit already told us a lot about
the proper treatment of EFTs also in the context of higher-order corrections,
i.e. not only at LO.
(iii) I feel that in the discussion in the paragraph before Eq. (5.15) the
treatment of the axion is not included in this setup. The axion is light, but
develops a wee coupling to SM particles. This cannot be treated within the
conventional EFTs. I find it worth to mention, since axions belong to
present-day's BSM physics. This might be mentioned in terms of a footnote or
so.
(iv) The RGEs of SMEFT are also partially known at NLL. May be the authors
would also like to refer to these works - mainly QCD. There are some
higher-order calculations within SMEFT for the LHC, i.e. beyond LO for the
SMEFT contributions.
(v) I would have expected much more original references for [37-39].
(vi) For me an important typo: The last Eq. of Eq. (5.24) must be an equation
for M_pi^2, not M_pi. The squared pion-mass - linear quark-mass relation is
very crucial. May be the authors would like to mention that their coefficient
B is related to the quark condensate? Or what is the reason to omit this
correspondence? Isn't it appropriate to cite the original and famous
Gell-Mann/Oakes/Renner paper for this?
(vii) Why don't the authors omit all NLO papers from Refs. [54-57] with the
chance to bring the attention of the reader to the real and explicit
shortcomings of previous works? In the way written, it appears to be very
vague.
(viii) Related to Eq. (5.32) I am missing an explicit definition of g and (thus
related) a proper settling of the correction Delta r.
(ix) I am puzzled by the notion that hadronic uncertainties only matter at NNLO
or beyond for the muon lifetime. The lifetime of the muon is affected by
hadronic effects (via Delta r) already at NLO. Or do I miss something? I think
that in succession to the previous point a rigorous clarification of the input
used is mandatory. Otherwise a discussion about how to use hadronic
contributions to Delta r as pursued in elw. precision fits would be useless...
(x) Is there any chance to use inelastic Compton scattering e gamma -> mu ->
e gamma for the mu-e-gamma coupling? This will of course be an issue
concerning errors on the exp. side. But it will be a resonant process. May be
as precisely as the decay process measurable - or not?
(xi) typos: quantum-field theoretic -> quantum-field theoretical
transfom -> transform
form. The authors provide a comprehensive theoretical introduction to PSI
experiments. However, there are a few (minor) issues that require some
modifications.
(i) I see that Refs. [1-23] are not final. I am expecting these citations to
be updated before this contribution will be published.
(ii) Related to Eq. (5.12) it might be worth to refer to the LE-Fermi limit of
the SM, i.e. that the Wilson coefficients of these operators may even acquire
higher-order corrections if matched to a particular model (as the SM). I am
emphasizing this very particular issue, since these matching conditions gave a
lot of information about the top mass and Higgs mass (after knowing the top
mass). I think that the SM in the low-energy limit already told us a lot about
the proper treatment of EFTs also in the context of higher-order corrections,
i.e. not only at LO.
(iii) I feel that in the discussion in the paragraph before Eq. (5.15) the
treatment of the axion is not included in this setup. The axion is light, but
develops a wee coupling to SM particles. This cannot be treated within the
conventional EFTs. I find it worth to mention, since axions belong to
present-day's BSM physics. This might be mentioned in terms of a footnote or
so.
(iv) The RGEs of SMEFT are also partially known at NLL. May be the authors
would also like to refer to these works - mainly QCD. There are some
higher-order calculations within SMEFT for the LHC, i.e. beyond LO for the
SMEFT contributions.
(v) I would have expected much more original references for [37-39].
(vi) For me an important typo: The last Eq. of Eq. (5.24) must be an equation
for M_pi^2, not M_pi. The squared pion-mass - linear quark-mass relation is
very crucial. May be the authors would like to mention that their coefficient
B is related to the quark condensate? Or what is the reason to omit this
correspondence? Isn't it appropriate to cite the original and famous
Gell-Mann/Oakes/Renner paper for this?
(vii) Why don't the authors omit all NLO papers from Refs. [54-57] with the
chance to bring the attention of the reader to the real and explicit
shortcomings of previous works? In the way written, it appears to be very
vague.
(viii) Related to Eq. (5.32) I am missing an explicit definition of g and (thus
related) a proper settling of the correction Delta r.
(ix) I am puzzled by the notion that hadronic uncertainties only matter at NNLO
or beyond for the muon lifetime. The lifetime of the muon is affected by
hadronic effects (via Delta r) already at NLO. Or do I miss something? I think
that in succession to the previous point a rigorous clarification of the input
used is mandatory. Otherwise a discussion about how to use hadronic
contributions to Delta r as pursued in elw. precision fits would be useless...
(x) Is there any chance to use inelastic Compton scattering e gamma -> mu ->
e gamma for the mu-e-gamma coupling? This will of course be an issue
concerning errors on the exp. side. But it will be a resonant process. May be
as precisely as the decay process measurable - or not?
(xi) typos: quantum-field theoretic -> quantum-field theoretical
transfom -> transform

---

## Round 1 · Referee Report · Anonymous (Referee 2) · 2021-7-6

Report
This is a very useful introduction to the theory tools necessary to the interpret experimental activities carried out at PSI. I think the paper is complete and well-organised.
My only concern is that section 5.2 is rather long and technical, and the article lacks of a vision/motivation part, especially as far as BSM measurements are concerned.
If the article is addressed to a broad audience (i.e. also to non EFT practitioners), as I assume, then I would recommend to extend a bit the introduction with a first (non-technical) clarification/exemplification of what is meant by BSM searches vs. precise determination of SM parameters vs. “auxiliary” measurements (in the context of PSI experiments). Most important, some discussion about the need for these 3 lines of research, in particular the theoretical motivations for BSM searches via precision measurements, and the distinction between light and heavy new physics, would also be very welcome if made in the introduction.
A more detailed structure of 5.2 with subsections would also be welcome.
My only concern is that section 5.2 is rather long and technical, and the article lacks of a vision/motivation part, especially as far as BSM measurements are concerned.
If the article is addressed to a broad audience (i.e. also to non EFT practitioners), as I assume, then I would recommend to extend a bit the introduction with a first (non-technical) clarification/exemplification of what is meant by BSM searches vs. precise determination of SM parameters vs. “auxiliary” measurements (in the context of PSI experiments). Most important, some discussion about the need for these 3 lines of research, in particular the theoretical motivations for BSM searches via precision measurements, and the distinction between light and heavy new physics, would also be very welcome if made in the introduction.
A more detailed structure of 5.2 with subsections would also be welcome.

---

## Round 2 · Referee Report · Anonymous · 2021-7-14

Report

With the modifications performed by the authors I am happy with the present version of the manuscript and recommend publication in SciPost.

---

## Round 2 · Referee Report · Anonymous · 2021-7-19

Report

My previous concerns have been well addressed.

---

## Round 2 · Author Response

We thank both referees for their comments and suggestions. They have triggered a list of changes given below.

---

## Round 2 · List of Changes

REPORT 1
========

Here we list our reply to the various points of Referee 1 and indicate the changes we have made. Equation numbers and line numbers refer to the revised version. For clarity and better connection to the referee's comments, the old numbers are sometimes given as {old NR}.

(i) Yes, Ref[1-23] will be finalized during the proof process, once the detailed information is available.

(ii) We fully agree with this remark but would like to point out that this has been mentioned after (5.35) where we state that the matching in (5.33) or (5.11) and (5.12) is done in the limit \delta r -> 0 (Line 336).

(iii) We have restructured Section 5.2 and at the beginning of Section 5.2.2. introduced a paragraph where we mention low-mass BSM and briefly discuss the axion / ALPs.

(iv) Yes, some partial results are known beyond LL. Our original formulation was indeed somewhat misleading and we have adapted this,
see Lines 180-182. We refrain from adding further references as it is impossible to do justice to the vast literature within our limited
scope.

(v) Indeed, a huge number of original articles were not explicitly mentioned in connection with [41-43] {old 37-39} and in fact many
other places as well. Given the space constraints we have used review articles whenever possible and invite the reader to consult those for further references. Otherwise the number of references would get out of hand.

(vi) We completely agree with the referee and thank him/her for pointing this out. The discussion has been adapted and extended, see
(5.23) - (5.27).

(vii) We are not entirely sure we understand the referee's point here. Refs [59-62] {old 54-57} are all NNLO calculations and we have
omitted all NLO calculations (and calculations in the logarithmic log[mmu/me] approximation) to keep the number of references under
control. We have added a remark (Line 333) to stress the importance of the NNLO QED calculation.

(viii) We have given a second version of (5.35) {old 5.32} where we state the use of the on-shell scheme. We also add a few remarks about the structure of \delta r (Lines 330-332). But a more precise definition of \delta r does in our view not belong into a specific PSI
theory discussion, even though it is of course of utmost importance.

(ix) Apparently our text was not clear enough and we have edited this part (Lines 310 - 343). The statement is that \delta q obtains
hadronic corrections only at NNLO. What the referee has in mind is that \delta r obtains hadronic corrections at NLO. This is true of
course and usually dealt with through the renormalization of \alpha (now mentioned in Line 331). Once more, we would like to point out
that we do not want to discuss the details of electroweak precision fits, as we feel they are not part of theory for PSI experiments. But
we hope with the current formulation the misunderstanding has been cleared up.

(x) This is an interesting question but we feel it is not really part of our article.

(xi) These typos have been corrected.

REPORT 2
========

We gladly take up the two recommendations of Referee 2.

In particular, we have restructured Section 5.2, starting with a brief outline and then structure the section into 3 subsections. In
addition, at the beginning of Section 5.2.2 we have included a brief discussion of low-mass BSM.

Finally, we have extended the introduction a bit and have given illustrative examples of experiments at the intensity frontier and
precision frontier.

---

## Editorial Decision

published